# Experimental Evaluation of Glulam Made from Portuguese Eucalyptus

**Aiuba Suleimana [1,2]**, **Bárbara C. Peixoto [3]**, **Jorge M. Branco [1,]*** and **Aires Camões [4]**

1. Department of Civil Engineering, University of Minho, ISISE, 4800-058 Guimarães, Portugal; asuleimana@unilurio.ac.mz
2. Rural Engineering Department, Lúrio University, Sanga 3302, Mozambique
3. School of Architecture, Art and Design, University of Minho, 4800-058 Guimarães, Portugal
4. Centre for Territory, Environment and Construction (CTAC), University of Minho, 4800-058 Guimarães, Portugal; aires@civil.uminho.pt
* Correspondence: jbranco@civil.uminho.pt

**Abstract:** Engineered wood products (EWPs) have evolved over time to become a popular and sustainable alternative to traditional lumber by offering design flexibility, increased strength, and improved quality control. This work analyzes the potential of Portuguese eucalyptus wood (*Eucalyptus globulus*) to produce glued-laminated timber (glulam) for structural applications. Currently, this hardwood is used for less noble applications in Portugal's construction industry. To promote the use of this species of timber in construction, an experimental campaign was conducted to characterize its compression parallel to the grain and bending strength. The results demonstrated that this hardwood presents a compression parallel to the grain strength of 73 N/mm$^2$ and a bending strength of 151 N/mm$^2$ with a global value of elastic modulus equal to 24,180 N/mm$^2$. Based on those strength values obtained from the glulam produced with eucalyptus, one can conclude that the test results presented here are higher than the ones declared by the current glulam made of softwoods; thus, additional studies are encouraged.

**Keywords:** glulam; hardwoods; eucalyptus; experimental evaluation





## 1. Introduction

The Portuguese Institute of Nature Conservation and Forests (ICNF) shows that the *Eucalyptus globulus* occupies 26.2% of all existing forests in the country, making it, already, the most representative wood species in Portugal [1]. This hardwood is originally from Australia and was introduced in Portugal in 1954 [2]. However, due to the fast-growing nature of the tree, with short harvesting times between the ages of 10 and 15 years, compared to the most representative local species, such as maritime pine (35 to 45 years) [3], in addition to its easy adaptation to soil and atmospheric conditions, made its introduction in the Portuguese forest easy and fast. Moreover, the growth of the national paper industry made it economically interesting for private proprietors to plant eucalyptus.

In Portugal, eucalyptus is primarily used for pulp and paper production, as well as for biomass as a source of energy [4]. In construction, this species has been used for parquet flooring for residential buildings, and for traditional applications, such as railway sleepers, mine structures [5], and for sports purposes, due to its hardness and clear color. In the past, in particular in rural regions, as a consequence of its availability and low cost, it has been used for structures such as roofs, floors, and walls. For example, the historical center of Guimarães is well known for its half-frame timber construction system of using eucalyptus wood since the second half of the 20th century.

The production of glued-laminated timber (glulam) normally uses softwoods such as spruce, fir, and Scots pine, which are the most common species in Central Europe and are where the most important manufacturers of glulam are located. By contrast, the



Portuguese forests present a high percentage of hardwoods, with eucalyptus being the most representative species. In this context, it will be interesting to study the possibility of using eucalyptus to produce glulam, thereby adding value to the Portuguese forests. It is possible to find German producers of glulam, which is made of beech from the black forest; however, to our knowledge, the only producers of glulam made from eucalyptus are located in Galicia, Spain, where the presence of this species is also important. Eucalyptus seeds were first introduced in Spain in the 1800s [6], and their woods have been used in different areas, from mercenaries to carpentry and buildings.

Most of the glulam is produced using softwoods; however, the use of some hardwoods was previously mentioned in EN 14080 [7]. Unfortunately, *Eucalyptus globulus* was not mentioned by this standard as a possible species to use in the production of glulam.

Seng Hua et al. [8] presented a summary of studies conducted on the different uses of eucalyptus species worldwide, with special emphasis on engineered wood products (EWP), such as glulam, cross-laminated timber (CLT), fiberboards, oriented strand boards (OSBs), laminated veneer timber (LVT), and other particleboards and plywood. It was demonstrated that glulam was the application least studied.

In Brazil, glulam beams made from *Eucalyptus urogandis* have been tested and obtained values for bending strength in the range of 57–94 $N/mm^2$, with an elastic modulus in the range of 17,800–20,100 $N/mm^2$, a compression strength parallel to the grain between 65 and 75 $N/mm^2$, and a bond line shear strength of 10–12 $N/mm^2$ [9], which suggested that the species had great potential to be applied in glulam industries owing to its high mechanical properties.

Despite this lack of information, Franke and Marto [10] evaluated the use of *Eucalyptus globulus* as an engineered material, achieving characteristic values of 115 $N/mm^2$ and 83 $N/mm^2$ for bending and compressive strength parallel to the grain, respectively. Moreover, the gluing and delamination properties were measured with PUR adhesive, and it was observed that none of the tried configurations were successful in meeting all the requirements regarding the strength and durability of the glue lines. Nevertheless, glue lines based on MUF adhesive were studied by Suleimana et al. [11], who obtained failures that occurred mostly on the wood side and achieved shear strengths of 14 $N/mm^2$ and 12 $N/mm^2$ using surface-bonding and edge-bonding specimens, respectively.

In Spain, solid and glulam elements made of *Eucalyptus globulus* have been evaluated and characterized by Lara-Bocanegra et al. [12], for use in timber grid shells. A bending strength of 55 $N/mm^2$ was achieved by the solid laths, while a strength class higher than a GL56 could be attributed to the glulam elements. In Portugal, homogeneous glulam beams made of *Eucalyptus globulus* and hybrid (eucalyptus and poplar) were evaluated by Martins et al. [13]. The average values obtained for bending modulus were 23,487–25,615 $N/mm^2$ and 18,302–22,341 $N/mm^2$ for local and global values, respectively, while the bending strength ranged from 91 to 115 $N/mm^2$ for the hybrid and homogenous glulam beams, respectively.

In the current work, an experimental evaluation of glulam beams made from *Eucalyptus globulus* is presented. Glulam beams were manufactured in an industrial facility and tested in the laboratory under four-point bending tests. Due to the lack of data about the mechanical characterization of the eucalyptus wood, small specimens, representing the lamellae used in the glulam beams, were used to prepare compression parallel to the grain and bending tests. The objective was to assess the mechanical characterization of the raw material used in the lamellae. The National Laboratory of Civil Engineering (LNEC) published results of an experimental campaign based on small clear samples of *Eucalyptus globulus* [14], which were used as references. Density values of 750–850 $kg/m^3$, bending strength of 128 $N/mm^2$, and a modulus of elasticity in bending of 17,500 $N/mm^2$ were demonstrated, while compression and tension strengths parallel to the grain were equal to 49 $N/mm^2$ and 14 $N/mm^2$, respectively, alongside a shear strength of 3 $N/mm^2$.

The main objective of the present work was to assess the possibility of producing glulam from *Eucalyptus globulus* grown in Portugal, to add value to this hardwood species, while reducing the national dependence on imported softwoods.

## 2. Materials and Methods

The eucalyptus specimens were collected from logs cut in north Portugal by a sawmill located close to Amarante. After their preparation and transport to laboratories at the University of Minho, all specimens were kept in a climatic chamber under a controlled temperature of 20 °C and relative humidity (RH) of 60% for approximately 4 weeks, until mass stabilization was reached, as recommended by NP 614 and ISO 3130 [15,16]. The lamellae for the glulam beam production were prepared in the *Rusticasa* industry facilities and the tests were conducted in the Laboratories of the Civil Engineering Department at the University of Minho. Table 1 summarizes the experimental campaign performed with the identified corresponding standards as follows.

**Table 1.** Summary of the experimental campaign performed.

| Property | Symbol | Unity | Standard |
| --- | --- | --- | --- |
| Moisture content | $\omega$ | % | ISO 3130 [16] |
| Density | $\rho_k$ | kg/m$^3$ | EN 13183-1 [17] |
| Compression parallel to the grain (clear specimens) | $f_{c,0}$ | N/mm$^2$ | ASTM D143 [18] |
| Bending strength (clear specimens) | $f_{m,90}$ | N/mm$^2$ | ASTM D143 [18] |
| Bending strength on structural beams | $f_{m,90}$ | N/mm$^2$ | EN 408 [19] |

### 2.1. Determination of Moisture Content and Density

After the stabilization of the specimens inside the climatic chamber, they were removed, and the dimensions and weight were measured to determine the density ($\rho$). Afterward, the tested specimens were conditioned in an oven to dry and the moisture content ($\omega$) was measured. Both characteristics of the specimens were determined by equations that can be found in ISO 3130 and EN 13183-1, respectively [16,17].

### 2.2. Compression Parallel to the Grain on Clear Specimens

The purpose of this test was to evaluate the compression strength parallel to the grain in clear specimens by following the guidelines in ASTM D143 [18]; the specimens were produced by following NP 618 [20]. These results allowed a control quality to be performed on the raw material of *Eucalyptus globulus* lamellae. Moreover, it was, then, possible to compare the results to the values suggested by LNEC [14].

The specimen was placed on the bottom plate of the machine centered on the vertical axis, as shown in Figure 1a,b. The actuator was lowered until it touches the specimen's face. Then, the load, registered by a load cell with a maximum capacity of 200 kN, was applied at a rate of 0.01 mm/second until the specimen was broken, which normally happened after 3 (+/−2) min.

To determine the results of compression strength parallel to the grain, Equation (1) was applied.

$$f_{c,0} = \frac{F_{max}}{bh} \tag{1}$$

where $f_{c,0}$, is the compression strength parallel to the grain in N/mm$^2$, $F_{max}$ is the maximum load in Newton, and *b* and *h* are the width and height of the specimen cross-section, respectively, measured in millimeters. In total, fourteen clear specimens were tested.

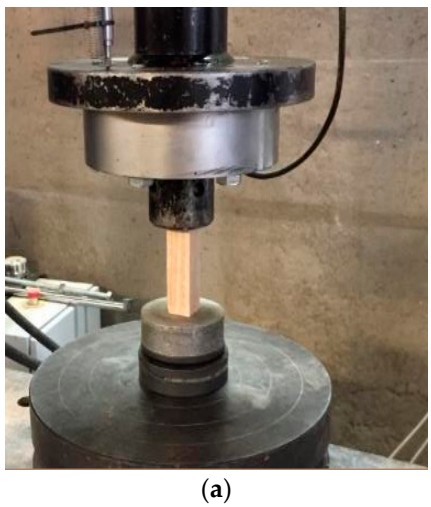

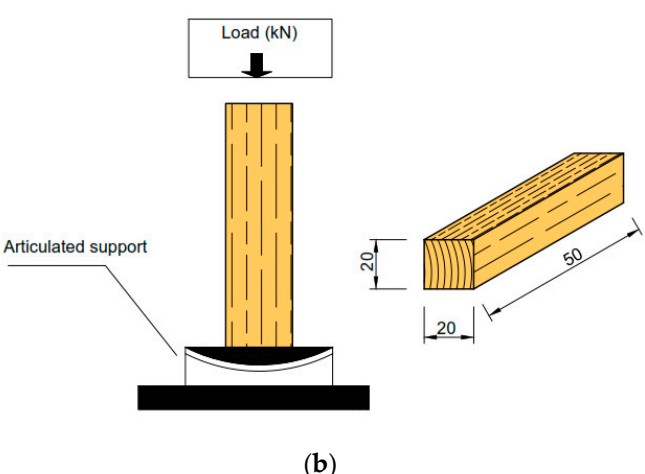

(**a**)

(**b**)

**Figure 1.** Compression parallel to the grain. (**a**) Test setup, (**b**) specimen.

### 2.3. Bending Strength on Clear Specimens

For the bending strength in clear specimens, tests according to NP 619 [21] and ASTM D143 [18] were performed. The machine's actuator was lowered until it touched the specimen's face. Then, a maximum actuator force of 25 kN with a constant speed of 0.01 mm/second was applied to ensure that the rupture occurred over a maximum period of 3 (+/−2) min. Figure 2a,b shows a representative bending failure and the setup used.

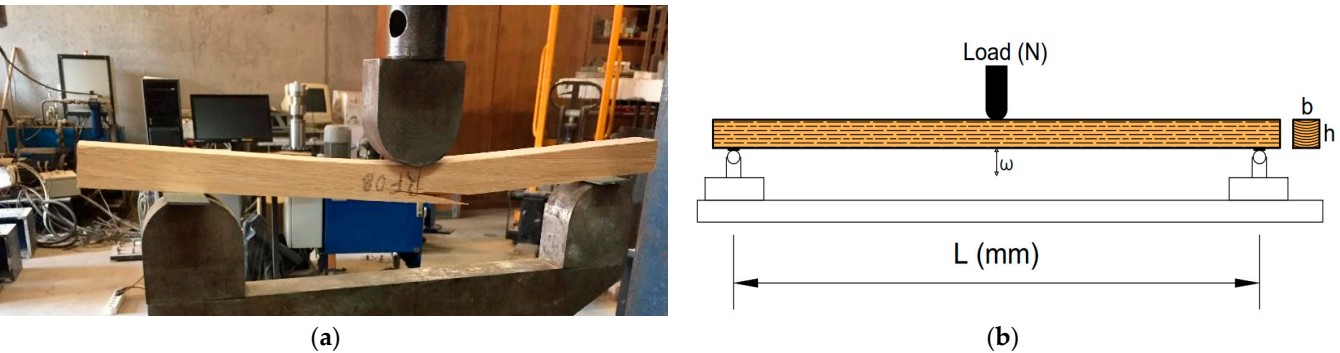

(**a**)

(**b**)

**Figure 2.** Configuration of four-point bending test according to NP 619 [21] and ASTM D143 [18]; (**a**) bending failure, (**b**) setup.

To determine the results of bending strength on the clear specimens, Equation (2) was applied, whereas to obtain the values of the moisture content and density, measurements for the weight and volume methods already described were applied to the specimens after testing the bending.

$$f_m = \frac{3Fl}{2bh^2} \left[ \text{N/mm}^2 \right] \qquad (2)$$

where $f_m$ is the static bending tensile strength, expressed in N/mm$^2$, $F$ is the maximum bending force recorded during the test, in Newton, $l$ is the span between the beam supports in millimeters, and $b$ and $h$ are the width and height of the specimen cross-section, respectively, in millimeter. In total, fifteen clear specimens were tested.

### 2.4. Production of the Glulam Beams

For the gluing of the glulam beams, a two-component of glue, so-called melamine-urea-formaldehyde MUF 1242/2542, produced by Akzo Nobel, was used (Table 2).

**Table 2.** MUF 1247/2526 technical specifications [22].

| Component | 1247 | | | 2526 | | |
|---|---|---|---|---|---|---|
| Product | MUF glue | | | Hardener | | |
| Appearance | Liquid | | | Liquid | | |
| Color | Opaque/White | | | White | | |
| Viscosity (At production moment) | 10,000–25,000 mPas (Brookfield LVT, sp.4, 12 rpm, 25 °C/ 77 °F) | | | 1700–2700 mPas (Brookfield LVT, sp.4, 60 rpm, 25 °C/ 77 °F) | | |
| PH (At production moment) | 9.5–10.7 (at 25 °C/77 °F) | | | 1.3–2.0 (at 25 °C/77 °F) | | |
| Dry layer | 64–69% | | | Not applicable | | |
| Storage time (months) | 15 °C/59 °F 4 | 20 °C/68 °F 4 | 30 °C/86 °F 2 | 15 °C/59 °F 4 | 20 °C/68 °F 4 | 30 °C/86 °F 2.5 |
| Information on formaldehyde | Free formaldehyde $\leq$ 0.8% | | | Formaldehyde free | | |
| Density | Approx. 1270 kg/m$^3$ | | | Approx. 1070 kg/m$^3$ | | |
| Properties of the glue | Lightly colored glue joints. High resistance to water and adverse environmental conditions. Meets the requirements of EN 301 [23] (for glue type I and II, service class 1, 2, and 3), EN 391 [24], EN 392 [25], and DIN 68141 [26]. | | | | | |

The production process began by sorting the boards and defining the beam configuration. The process of beam manufacturing was conducted by the *Rusticasa* Company as described below.

Firstly, the eucalyptus wood was obtained, and then, cut to produce lamellas of 2500 mm in length and with a transversal cross-section of 76 × 21 mm$^2$. After that, the lamellas were classified visually to make a control and remove the most defective lamellas; the approved ones are shown in Figure 1a. Then, the approved lamellas were kept in a climatic chamber for three months at a temperature of 20 °C and a relative humidity of 60%, for the glue curing to occur. Periodic measurement was performed to control moisture stabilization. After drying, the lamellas were removed from the climatic chamber and planed on the same day as the glue was applied (less than 6 h).

With the glue applicator on, as shown in Figure 3b, one person holds the first lamellae and passes it from the middle of the glue applicator to another person and the second one applies it to the press bed. This process provides a possibility for the uniform application of the glue and was performed for seven lamellae before it was interrupted. To separate between beams, the lamellae, without glue, were placed on the top of the group and the same process of passing the lamellas from the middle of the glue applicator was repeated. This step required 20 min to be completed for all the beams.

After all the lamellas were glued and placed on the press bed, they were subjected to a pressure of between 133 bar and 135 bar, for one hour, for the gluing and curing to occur, as shown in Figure 1c. Thereafter, they were removed from the press, the remaining glue was cleaned and the faces and the corners were adjusted for the dimensions, shown in Figure 1d. Finally, they were taken to the Laboratories of the Civil Engineering Department of the University of Minho for testing. Note that, the same procedure mentioned above was followed by Martins et al. [27] and Pedro et al. [28] in the production of glulam using other hardwoods (*Acacia melanoxylon* and *Acacia mangium*, respectively).

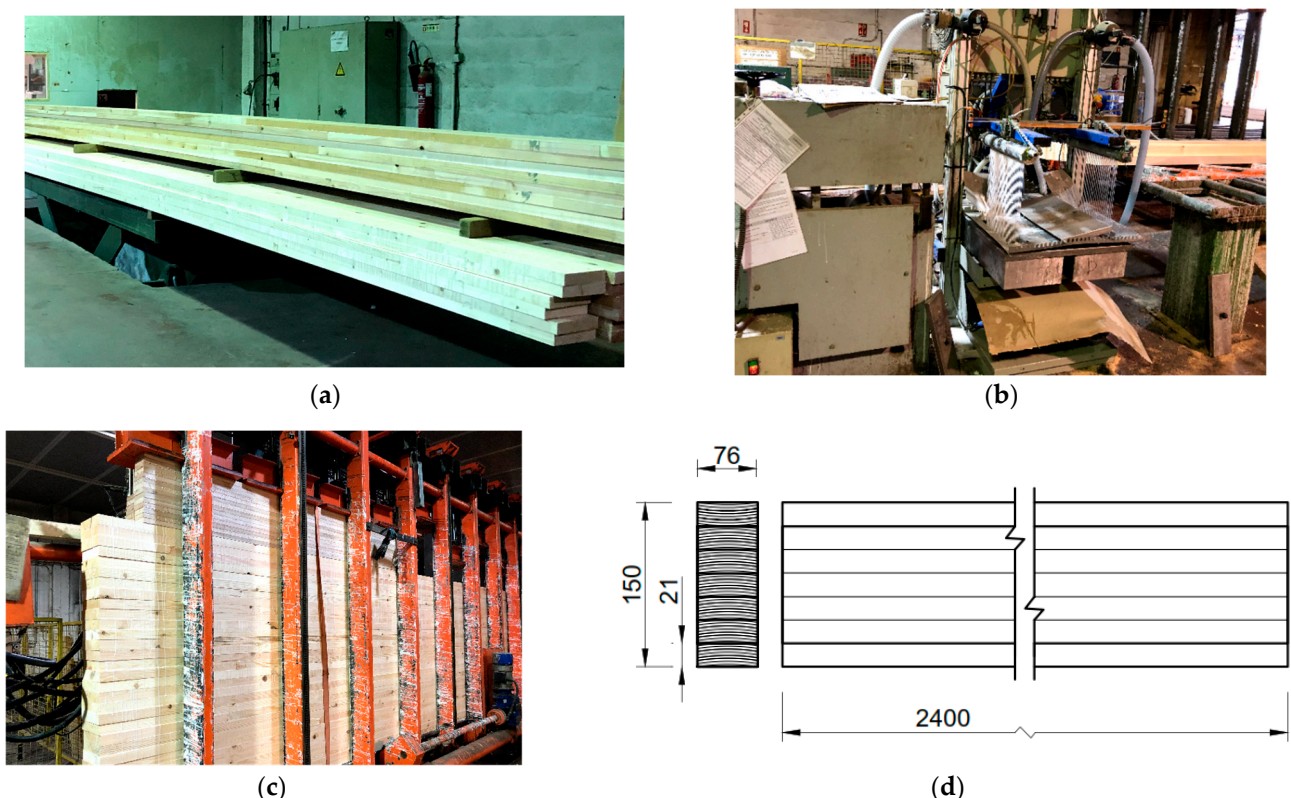

**Figure 3.** Different stages of the industrial process to produce glulam and the configuration of beams; (**a**) paired boards ready for glue application, (**b**) glue machine in use, (**c**) boards with glue stacked and applying pressure for gluing, (**d**) configuration of beams in millimeters.

### 2.5. Bending Tests on Beams

The bending tests performed on structural glulam beams produced with eucalyptus aimed to quantify their main bending properties, namely, global values of the modulus of elasticity in bending ($E_{m,g}$) and the corresponding bending strength ($f_m$). These mechanical properties were measured following the recommendation made by EN 408 [19] with regard to the specimen's geometry, test, setup, loading protocol, and instrumentation. Figure 4 illustrates the four-point bending test on a glulam beam made from eucalyptus. In total, four glulam beams were tested.

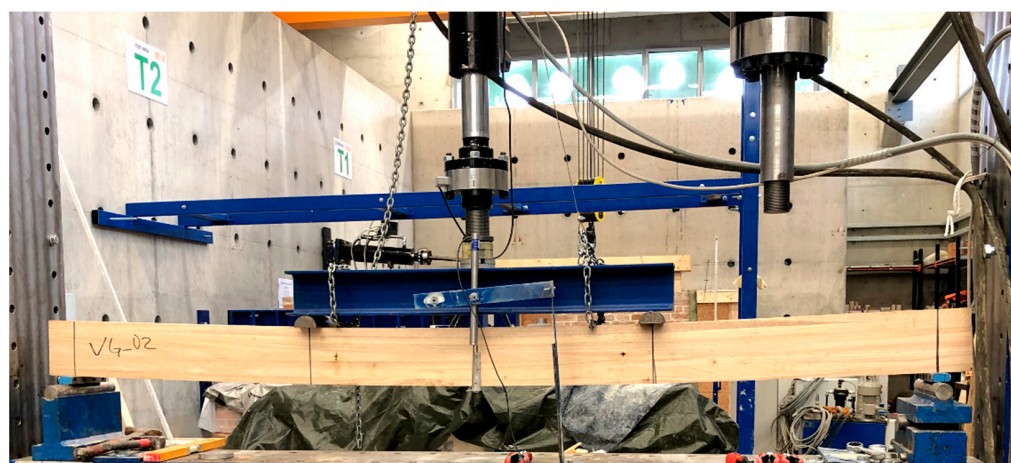

**Figure 4.** Four-point bending test on a glulam beam.

Bending strength of the beams is determined by Equation (3):

$$f_m = \frac{3aF_{max}}{bh^2} \ [\text{N/mm}^2] \tag{3}$$

where $F_{max}$ is the maximum load applied in Newton, $b$ and $h$ are the width and height of the beam's cross-section, respectively, in millimeters, and $a$ is the distance between the point where the load is applied and its closest support.

In addition, the global value of the elastic modulus in bending is obtained by Equation (4).

$$E_{m,g} = \frac{l^3 (F_2 - F_1)}{bh^3 (w_2 - w_1)} \left[ \left( \frac{3a}{4l} \right) - \left( \frac{a}{l} \right)^3 \right] \ [\text{N/mm}^2] \tag{4}$$

where the $F_1$ and $F_2$ are the loadings corresponding to 10% and 40% of the maximum load, respectively, in Newton, $w_1$ and $w_2$ are the displacements corresponding to loads of $F_1$ and $F_2$, respectively, in millimeters, $l$ is the distance between the two supports in millimeters, and $b$, $h$ and $a$, again, assume the meaning already presented for Equation (3).

## 3. Results

Here, all test results are presented and, when possible, comparisons with existing values are made.

### 3.1. Results of Clear Specimens

Density ($\rho$) and moisture content ($\omega$) of all specimens, as already mentioned, were quantified according to ISO 3130 [16] and EN 13183-1 [17], respectively. In the first phase, the density was calculated considering the boards with their initial dimensions before the bending tests. Subsequently, the density was measured after the mechanical tests near the failure zones for both structural and smaller specimens. The results obtained are summarized in Table 3. The results of the mechanical tests performed on small clear specimens for bending and compression, parallel to the grain, are depicted in Figure 5. The failures obtained were, as expected, tension and crushing, respectively, which are presented in Figure 5c,d.

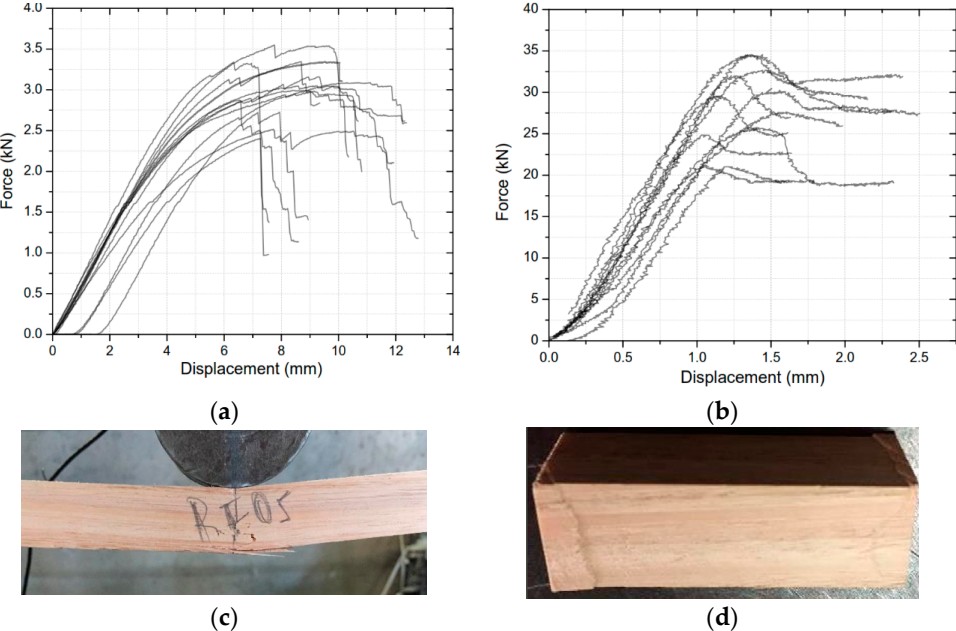

**Figure 5.** Experimental load-displacement curves obtained from the bending (**a**) and compression parallel to the grain (**b**) tests, and respective failure mode associated, tension (**c**) under bending and crush (**d**) under compression parallel to the grain.

**Table 3.** Results of physical and mechanical characterization on clear specimens.

| Characteristic | | $\rho$ (kg/m$^3$) | $\omega$ (%) | $f_{c,0}$ (N/mm$^2$) | $f_m$ (N/mm$^2$) |
|---|---|---|---|---|---|
| Compression parallel to the grain | Mean | 802.43 | 9.65 | 72.84 | |
| | Maximum | 845.42 | 10.87 | 86.35 | |
| | Minimum | 691.53 | 8.74 | 52.5 | |
| | CoV (%) | 8.51 | 6.79 | 13.25% | |
| Bending | Mean | 809.5 | 8.86 | | 151.34 |
| | Maximum | 958.21 | 10.19 | | 175.88 |
| | Minimum | 615 | 6.58 | | 118.65 |
| | CoV (%) | 12.33 | 9.24 | | 11.66% |

Table 3 presents the average results obtained from the compression parallel to the grain and bending tests, corresponding to the 14 and 15 tested specimens, respectively, as well as its maximum, minimum, and coefficient of variation, CoV (%).

The results obtained for the clear specimens show similar values of strength for bending ($f_m$) and compression, parallel to the grain ($f_{c,0}$), which were calculated according to NP 618 [20] and NP 619 [21], respectively, and were also conciliated and supported by ASTM D143 [18], which correspond to averages of 72.84 N/mm$^2$ and 151.34 N/mm$^2$, respectively. The coefficient of variation was minus or equal to 13.26%. It is important to note that the observed failure on the clear specimens was because of tension under the bending strength (Figure 5c) and by crush under compression strength, parallel to the grain (Figure 5d). Moreover, the density ranged between 802 kg/m$^3$ and 813 kg/m$^3$, with a CoV of 12.5%, while the moisture content varied between 9% and 11%.

### 3.2. Bending Tests on Glulam Beams

The bending test results obtained for glulam beams produced in this study with Portuguese eucalyptus are presented in Table 4.

**Table 4.** Results of the bending tests on the glulam beams produced from eucalyptus.

| | $\rho$ (kg/m$^3$) | $\omega$ (%) | $f_m$ (N/mm$^2$) | $E_{mg}$ (N/mm$^2$) |
|---|---|---|---|---|
| Mean | 812.70 | 11.07 | 96.74 | 24,180 |
| Maximum | 854.83 | 11.65 | 106.16 | 26,300 |
| Minimum | 777.19 | 10.79 | 89.36 | 22,580 |
| CoV (%) | 3.68 | 3.13 | 7.37 | 6.46 |

The glulam beams made from eucalyptus, on average, have a density of 812.70 kg/m$^3$ and a bending strength of 96.74 N/mm$^2$, which is a value three times bigger than the higher strength class for glulam made of softwoods, GL32 [7]. Moreover, the results obtained for the modulus of elasticity in bending ranges from 22,580 N/mm$^2$ to 26,300 N/mm$^2$, which are values approximately double the one of GL32 (32 N/mm$^2$ and 14,200 N/mm$^2$ for bending strength and global elasticity modulus, respectively). The force versus displacement curves are presented in Figure 6a and it is important to point out that all beam tests have reached failure modes typical for bending parallel to the grain, with a failure by pure tension and progress of the failure by shear (Figure 6b,c).

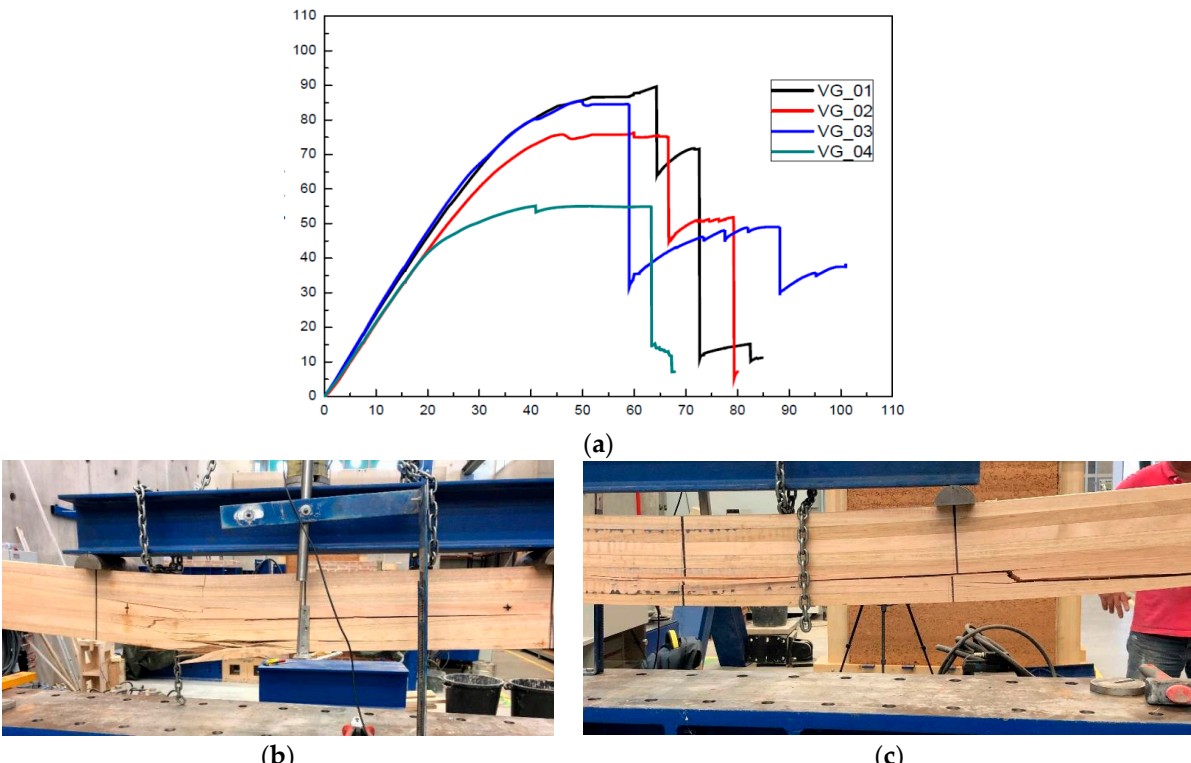

(**a**)

(**b**)                                                                   (**c**)

**Figure 6.** Curves of applied loads versus displacement (**a**). Typical failure modes of timber elements under bending. (**b**) Failure by pure tension and (**c**) failure progress by shear.

## 4. Discussion

Table 5 compares the results reported by the National Laboratory of Civil Engineering (LNEC) based on small clear specimens of *Eucalyptus globulus* and the average values obtained in the experimental campaign performed.

**Table 5.** Summary of the properties evaluated and reported by LNEC for the Portuguese *Eucalyptus globulus*.

| Properties | | Unity | LNEC [14] | Obtained Results |
|---|---|---|---|---|
| Density | | kg/m$^3$ | 750–850 | 803–813 |
| Bending strength | Module of Rupture | N/mm$^2$ | 127.5 | 151.3 |
| Compression parallel to the grain | Strength of failure | N/mm$^2$ | 49.1 | 72.8 |
| Bending strength (Glulam beams) | Module of Rupture Elasticity modulus | N/mm$^2$ | - - | 96.7 24,180 |

The specimens measured have an average density of 805 kg/m$^3$, corresponding to a moisture content of clear and small-sized, which was taken from glulam specimens less or equal to 12.5%, as recommended by EN 13183-1 [17]. These values are similar to the corresponding results from LNEC [14], although they are less than the one presented by Franke, and Marto [10] (980 kg/m$^3$) and higher than the one presented by Almeida et al. [29] (730 kg/m$^3$). Therefore, compared to results from LNEC and previous works, the values obtained in this experimental campaign can be assumed to be coherent and slightly better.

The compression parallel to the grain and bending strength attained average values of 73 N/mm$^2$ and 151 N/mm$^2$, respectively. These values are higher than the ones presented by LNEC [14], by 48% and 19%, respectively. Regarding the ones presented by Franke and Marto [10], similar values were achieved for the compression strength parallel to the grain

(83 N/mm$^2$, with a 14% difference) while it was higher than the bending strength value (115 N/mm$^2$). Moreover, this study pointed out experimental values higher than those presented by Acosta [30] for the compression strength parallel to the grain and bending strength, which were 89 N/mm$^2$ and 116 N/mm$^2$, respectively. This variety between the values presented in distinct studies can be justified by different parameters such as the origin and the age of the trees. It is worth highlighting that the average bending strength and global elastic modulus values of 97 N/mm$^2$ and 24,180 N/mm$^2$, respectively, registered for the glulam beams tested according to EN 408 [19], were higher than the maximum ones suggested by EN 14080 [7] for glulam elements made of softwoods. Moreover, they were higher than the ones obtained by an experimental campaign, conducted by Nogueira et al. [9], on full-scale glulam beams manufactured with *Eucalyptus urograndis* in Brazil. Nogueira et al. [9] achieved a bending strength in the range of 57–94 N/mm$^2$ and 17,812–20,072 N/mm$^2$ for the modulus of elasticity, with a compression strength parallel to the grain strength of 65–75 N/mm$^2$.

The results obtained in the experimental campaign performed demonstrate the promising mechanical properties of glulam made with *Eucalyptus globulus*, which can be considered a competitive EWP for structural applications and an excellent alternative to the current glulam made of softwoods.

## 5. Conclusions

The experimental campaign performed, and here presented, contributes to the hypothesis that *Eucalyptus globulus* from Portuguese forests can be used to produce glulam. Firstly, it was demonstrated that it is possible to produce glulam made from Portuguese eucalyptus in an industrial environment without any special provisions or changes to the regular production process, including gluing. Then, the obtained mechanical properties of the glulam made of eucalyptus were shown to be superior to the high strength class defined for current glulam made of softwood (GL32). Therefore, one can conclude that the use of this hardwood species, which is already the dominant species in Portugal, can contribute to adding value to Portuguese forests. It is important to note that the glulam produced with Portuguese *Eucalyptus globulus* exhibited mechanical properties higher than those found in the literature. However, only with a full and detailed characterization of this wood species, and the glulam elements produced, can their use in construction be considered. In particular, the assessment of the presence of finger-joints in the response to the glulam elements must be performed in future research since, in this study, no finger-joints were considered.

**Author Contributions:** Conceptualization, J.M.B. and A.C.; Methodology, A.S., B.C.P. and A.C.; Validation, J.M.B.; Investigation, A.S. and B.C.P.; Resources, J.M.B.; Writing–original draft, A.S.; Writing–review & editing, B.C.P., J.M.B. and A.C.; Supervision, A.C.; Funding acquisition, J.M.B. All authors have read and agreed to the published version of the manuscript.

**Funding:** This work was financed by Transform Agenda, approved under notice N°02/C05-i01/2022. Investment supported by the PRR-Recovery and Resilience Plan and by the NextGeneration EU European Funds and through a PhD grant SFRH/BD/151442/2021 conceded to the first author.

**Institutional Review Board Statement:** Not applicable.

**Informed Consent Statement:** Not applicable.

**Data Availability Statement:** Not applicable.

**Conflicts of Interest:** The authors declare no conflict of interest.

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
