# Peer review of "Experimental Evaluation of Glulam Made from Portuguese Eucalyptus"

_applsci, doi:10.3390/app13126866_

Round 1

Reviewer 1 Report

The manuscript is well written, and the topic is very interesting, only a few suggestions:

The abstract needs rewriting, please focus more on your results and implications for the industry and further research.

In the Materials and Methods please add a source of the wood in the research.

Lines 99-102 - no need to explain.

Lines 112-119: Please add only Standard, no need to add these formulas.

Please unify Glue/adhesive

In the Conclusions section please add Limitations of your research and Implications for further research.

minor changes needed

Author Response

1) the abstract has been rewrited with more focus on the results achieved and its implication in the next research steps.

2) The origin of the Eucalyptus wood was added

3) The formulas were removed and replaced by the reference to the standards

4) The use of adhesive was unify.

5) In the conclusions, the limitation of the research were added and next steps of the research are now discussed.

Reviewer 2 Report

Dear Authors,

Your study refers to the simple test of the mechanical properties of Eucalyptus globulus wood and glulam made of it to assess the possibility of producing glulam from Eucalyptus globulus grown in Portugal, but what about the possibility? What is the criterion value? When is it possible, and when not? Without clear criteria, the test makes no sense because it is not known what objectively the results should be related to. Otherwise, the conclusion is only speculation.

Please provide some keywords. They are crucial to understanding the concept of the article.

Specify in the introduction the standard requirements for the studied mechanical properties of glulam. They could create the criterion of eucalyptus glulam production possibility. You refer to the results of various studies by other authors, but it is unknown what reference values to refer to.

L. 212-214, is this text necessary? Table 3 is referred higher.

Be consistent when expressing units of properties N/mm2 or MPa. Also, use SI units everywhere.

Formulate the aim so that it clearly relates to what you want to achieve. This one is incomprehensible, inaccurate, non-specific, and impossible to achieve. Do you really want to reduce the national dependence of imported softwoods by mechanical tests of wood material?

Did you study the production line performance? I see - no. So, dont conclude about it.

The same applies to the economic assets. Be consistent - your study is only on the mechanical properties.

Author Response

1) the objective and aim of the study were clarified. First, the aim was to assess the possibility to produce glulam made of Eucalyptus. Then, a brief mechanical characterization, in particular under bending, was the goal. The abstract was upgraded to make it clear. Moreover, conclusions section has been also improved with the aim to make it clear the definition of the objectives of the work and the outcomes achieved.

2) The standard that defines the production of glulam is the EN 14080, as mentioned by the text.

3) Table has been improved.

4) Units have been made consistent.

5) we have kept the discussion of the outcomes to the glulam characterization, removing the attempt to extend some conclusions to economy.

Reviewer 3 Report

Dear Authors,

The article examines the potential of Portuguese Eucalyptus wood to produce glued laminated wood for structural purposes. An experimental campaign was carried out to evaluate the physical and mechanical properties of the wood. The results show that the mechanical properties of the hardwood are higher than those of softwood species, indicating its suitability for use in construction. The study suggests that using this wood species for structural applications is a promising option.

The title "Experimental evaluation of Glulam made from Portuguese Eucalyptus" is the accurate title for an article that discusses the experimental evaluation of glulam made from Portuguese Eucalyptus. It accurately reflects the article's focus and provides readers with an understanding of the article.

After reading the main text, I have some facultative comments for consideration by the authors:

1.       Line 25. Please verify the percentage. According to the Food and Agriculture Organization of the United Nations (FAO), the percentage of Eucalyptus globulus in forested land in Portugal was 26.5% in 2020 (https://www.fao.org/3/cb0048en/cb0048en.pdf), while before 2019 this percentage was 26% (https://www.mdpi.com/1999-4907/10/11/974). In my opinion, now it is at least 27%. Please refer most recent sources.

2.       Lines 33-39. Please supplement the areas of uses of the Eucalyptus globulus as biomass and in energy production. I suppose this is a second area of use of Eucalyptus globulus after being used as raw material in paper-like production. Additionally, Eucalyptus globulus is not commonly used as railway sleepers in Portugal. Pine and oak are the most commonly used wood species for railway sleepers in Portugal. Eucalyptus globulus is sometimes used as a furniture material in Portugal. The wood is known for its attractive grain and is used in various furniture applications, including chairs, tables, and cabinets. Its light color and relatively low density make it a popular choice for furniture makers. So I suggest mentioning energy use and furniture instead of railway sleepers.

3.       Line 100. "two times every two days a week"?, maybe it should be written "twice every two days"? Please rephrase.

4.       Below line 116 (eq. 2), please italicize the variables and the units in plain script.

5.       Line 123. Why authors used a 20×20×50 mm test samples, while Compression Perpendicular to Grain tests according to ASTM D143 use samples that are approximately 25.4 mm (1 inch) square in cross section and 2 to 4 inches long (25.4×25.4×50.8-76.2 mm). Why authors did not use EN 408. EN 408 does not provide specific sizes for the compression perpendicular to the grain test. Instead, it provides guidelines for preparing test specimens, which should be representative of the product and tested according to the relevant national standard (for example, samples of glulam beams used by authors).

6.       Line 153. Please expand the Table 2. MUF" in the table title. Additionally Table 2. I suggest changing the comma to a period in pH units. Additionally, "at 25 °C", not "a 25 °C).

7.       Lines 200-201. The sentence "Here, all test results are presented and analyzed and, when possible, a comparison with existing values is made." is redundant.

8.       Line 215. The title of Table 3, "Results of compression parallel to the grain and bending tests on small clear specimens", is confusing (or incomplete). The table also contains density and moisture content measurements. Additionally, I suggest providing the table columns titles.

9.       Line 275. I propose paraphrasing the study aim at the beginning of the "Conclusion" section.

Generally, the article's suggest that using Eucalyptus globulus from Portuguese forests to produce glulam is a promising solution to reduce dependence on glulam made of softwoods, as the mechanical properties obtained are superior to those typically presented by softwood species. Furthermore, the value added to this hardwood species could represent an important economic asset. The article also highlights that glulam made with Portuguese Eucalyptus globulus can be produced in an industrial environment without any special provisions or changes in the production line. The mechanical properties obtained are higher than those reported in previous literature. Overall, the conclusions appear informative and suggest potential benefits of using Eucalyptus globulus for glulam production.

I believe the article is interesting and meets the scientific articles standards.

Sincerely,

Author Response

  1. The FAO source is not clear about the 26.5% so we have referred to [1].

  2. The suggestion has been taken into account. Biomass use was added while the reference that demonstrates the use as railway sleepers and mines structures in Portugal, has been added.

  3. The sentence has been removed.
  4. The equation has been removed as suggested by other reviewer
  5. The idea was to have compression and bending tests on small specimens and therefore ASTM has been used. On the other hand, for the full-scale, or structural dimensions specimens, the EN 408 has been used.
  6. Added and updated.
  7. The sentence has been updated.
  8. The caption of Table 3 has been updated
  9. The suggestion of the reviewer was accepted.

Round 2

Reviewer 2 Report

Dear Authors,

please refer to my comments in the first round of reviews,

You did not precise the aim of the study. You still want to reduce the national dependence of imported softwoods. How was the reduction based on your results? Can you conclude this? This is the aim but no method, no results, no conclusion. So, the aim is wrong. You have to state the proper aim of the study.

What about the criteria of production possibility?

Author Response

Thank you for the positive input. In fact, the aim is not to reduce the dependency from softwoods. The aim is "just" to assess the possibility to produce glulam made of Eucalyptus and with that, to add value to the Portuguese forest. This works shows that this is possible (produce glulam made of eucalyptus) but further studies are necessary.
